# The Logopedic Evaluation of Adult Patients after Orthognathic Surgery

Anna Lichnowska *[ID] and Marcin Kozakiewicz [ID]

Department of Maxillofacial Surgery, Medical University of Lodz, 113th S. Żeromskiego, 90-549 Lodz, Poland; marcin.kozakiewicz@umed.lodz.pl
* Correspondence: anna.lichnowska@gmail.com

**Featured Application: The impact of the tongue and lips status in malocclusion is neglected in adults. Orthodontic and orthognathic treatment leads to dental and visual correction of the face but leaves a functional deficiency in the form of a speech disorder. This study highlights the essential role of speech therapy in adults. Evaluation and correction of primary functions along with the quality of speech and speech therapy should be a part of the team treatment of malocclusion and facial skeletal deformities.**

**Abstract:** Orthodontists correct dental malocclusion, but major facial skeleton deformations (skeletal malocclusion) are often subject to surgical correction. Several speech pathologies are associated with both of the occlusal anomalies mentioned above. The majority of articulation disorders and primary functions cannot be improved without skeletal correction. This study aimed to investigate the outcome of the multimodal and logopaedics treatment of Polish adults affected by skeletal malocclusion and speech-language pathology. A total of 37 adults affected by skeletal Class II and III malocclusion were included, along with the relationship between the malocclusion and speech deficiency (20 phonemes tested) in the subjects before and after surgical correction. The impact of surgery on pronunciation improvement and types of Polish phonemes most often misarticulated by Polish adults were also examined. Patients underwent combined treatment and received a full speech pathology examination. The treatment improved speech ($p < 0.05$), but the study did not prove that a specific surgery type was associated with pronunciation improvement. Some patients were provided with speech therapy during childhood, yet most had some minor difficulties with lip and tongue movements. Palatal, alveolar ($p < 0.05$), fricatives ($p < 0.05$), and labiodental consonant pronunciation ($p < 0.05$) improved. The surgical correction of malocclusion leads to better articulation of Polish consonants in adults and improves some primary functions.

**Keywords:** logopaedics; articulation; motor speech disorders; speech-language pathology; skeletal malocclusion; multimodal treatment; adults

## 1. Introduction

As an interdiscplinary science, speech pathology notably embraces general dentistry, orthodontics, maxillofacial surgery, and physiotherapy. As such, logopaedics becomes multidisciplinary. The results of studies on the relationship between articulation disorders and malocclusion have been contradictory, as many researchers, for more than 60 years, focused on skeletal deformities such as open bite. In open bite, speech disorders are mainly caused by abnormal tongue movements and atypical swallowing [1–3]. In 1980, Warren et al. [4] pointed out that speech disorders may be caused by other factors than open bite and proved that different physical or psychological factors might be involved in such cases. Blyth [5], Lubit [6], and Subtelny et al. [7] reported speech disorders accompanied by deep bite, Class II division I and Class III. In light of the worldwide literature, there is a high need to investigate the Polish language (Indo-European language group) in this field,

as it lacks well-designed and thoroughly conducted research. T. Laine reported several Finnish (Uralic language group) studies based on malocclusion traits and articulatory components of speech [8], articulatory disorders in speech related to the size of the alveolar arches [9] or position of the incisors [10], the relationship between interincisial occlusion and articulatory components of speech [11], and associations between articulatory disorders in speech and occlusal anomalies [12]. All of these studies were accompanied by an accurate speech examination performed by a qualified speech pathologist. L. Vallino, a clinical speech scientist, presented articles concerning English pronunciation (Indo-European language group) and velopharyngeal function before and after orthognathic surgery [13]. She promoted the statement that malocclusion requiring orthognathic surgery has a very negative impact on speech. In an article on perceptual characteristics of consonant errors associated with malocclusion [14], she supported her previous statement. She proved that some phonemes are more sensitive to deformation in dental and skeletal jaw relationships. D. Galvao de Almeida Prado et al. [15] presented an article on oral motor control and orofacial functions in the case of dentofacial deformity. These authors suggest that such deformities lead to imbalances in the stomatognathic system and affect the relationship between articulatory control and chewing, swallowing, and speech functions. This study concludes that weak oral motor control and disturbed primary functions cause significant instability for normative articulation. More detailed studies on the relationship between orthognathic surgery and speech were presented by two speech therapists, T.Guyette [16] and M.Wakumoto [17] (Indo-European language group). Guyette's considerations highlighted the impact of maxillary distraction osteogenesis, wheras Wakumoto's concentrated on fricative consonant pronunciation following osteotomy.

The human body does not have a separate organ for speech production. It is an element of the stomatognathic and digestive system; thus, disorders of these systems will always result in distortions of speech and the disruption of essential body functions necessary to sustain life, such as breathing, swallowing, and food intake, even though some compensations may be observed. Articulation is a coordinated set of peripheral movements of the speech organs and their systems while pronouncing sounds. Articulation disorders may either manifest in various syndromes characteristic of particular forms of speech pathologies or occur as isolated phenomena [18]. The occurrence of disorders in Polish phonemes articulation and the coexistence of malocclusions, such as open bite and various forms of prognathism or retrognathism, have restricted medical and speech pathology societies for a long time. No such research on adults in the Polish population has been published, apart from one study on the effectiveness of frenotomy in malocclusion and its impact on speech [19] and the doctoral thesis of L. Konopska [20], which was never presented worldwide. It would be interesting to investigate the speech and articulation outcome of the multimodal and logopaedic treatment of Polish adults affected by skeletal malocclusion and speech pathology. Although it belongs to Indo-European language group, Polish is a Slavic language, as opposed to English, which belongs to the group of Germanic languages (this determines the occurrence of regional voicing). Due to the lack of such research in the available literature, this issue seems essential, primarily if it is based on examining pronunciation before and after orthognathic surgery.

The aim of the study was the logopedic evaluation of patients after corrective surgery of skeletal malocclussion in Class II and III.

## 2. Materials and Methods

This study was approved by the Bioethical Committee of the University of Łódź no. RNN/73/19/KE. The study was based on the following aspects:

- relationship between malocclusion and speech defect in subjects before and after surgical correction;
- surgery impact on the improvement of pronunciation;
- types of Polish phonemes that are most often misarticulated by Polish adults;

- additionally, the data of all patients who reported to the Speech Therapy Outpatient Department were analysed for the relationship of skeletal malocclusion to phoneme pronunciation quality.

For the research, it was crucial to establish inclusion and exclusion criteria, presented in Table 1.

**Table 1.** Inclusion and exclusion criteria.

| Inclusion Criteria | Exclusion Criteria |
| --- | --- |
| (1) skeletal Class II or skeletal Class III | (1) cleft lip/palate |
| (2) age between 18 and 50 | (2) alveolar process cleft |
| (3) misarticulation | (3) treatment protocol: surgical correction of dysgnathia (surgery first), followed by orthodontic treatment |
| (4) patients' consent for surgery | (4) lack of any dysgnathia |
| (5) patients' consent for speech diagnosis | (5) lack of any of patients' consent or documents |
| (6) treatment protocol: orthodontic decompensation, followed by surgical correction of dysgnathia | (6) age under 18 and above 50 |
| (7) speech diagnosis before and after surgery | (7) hearing loss/disorder |

The research group consisted of 37 patients, including 26 females and 11 males, with an average age of $24 \pm 6$ years, who were patients of the Department of the Maxillofacial Surgery of Medical University of Łódź. The patients were enrolled in the research randomly, based on surgery date and the possibility of coordinating all medical and speech examinations before and after the surgery. All the selected participants had undergone combined orthodontic and orthognathic treatment accompanied by a speech pathologist's diagnosis, therapy (if necessary), and evaluation. Twenty-eight patients were diagnosed with some type of mandibular prognathism (skeletal malocclusion Class III: 19 prognathism cases and 9 laterognathism cases), and nine were diagnosed with some type of mandibular retrognathism (skeletal malocclusion Class II). Gender was not a defining criterion. The study was conducted in individual stages: a patient's qualification for combined treatment, along with orthodontic diagnosis, and an interview with a particular emphasis on information concerning previous speech therapy, along with external and intraoral examinations performed by a speech pathologist. The reference group was established as a negative reference (for comparison, there was a phonetic norm defined by the maximum test scores) consisting of patients without sufficient motivation to enter the multimodal treatment offered to them. There is no reference group to untreated patients at the institution from which this clinical material is presented, except for those who have opted out of treatment (and for whom it was only possible to perform an initial examination). The Outpatient Clinic does not provide standard speech therapy for adults due to patients not reporting there. Adult patients without malocclusion had been treated during childhood in Poland (or had refrained from speech therapy out of inner conviction). The reference group consisted of next 37 patients aged $25 \pm 4$ years, 21 females and 16 males, 19 with mandibular prognathism, 9 with mandibular laterognathism, and 9 with mandibular retrognathism. There was not a statistically significant difference amongst the age median in the treated patients group at a 95% confidence level (Kruskal–Wallis statistic = 0.7507 and $p$ = 0.3862).

The study also took into account overall language proficiency and thorough phonetic analysis of the collected speech samples. Speech pathology diagnosis was based on the Speech Organ Fitness Questionnaire [21] and the Articulation Test for Children and Adolescents [22]. Both tools were adapted for this study (the entire inventory of Polish vowels and consonants was not studied; thus, the ones indicated in worldwide studies as the most often misarticulated were selected).

The following phonemes were examined: /s/, /z/, /c/, /t/, /d/, /n/, /l/, /r/, /f/, /w/, /sz/, /ż/, /cz/, /dż/, /ś/, /ź/, /ć/, /dź/, /p/, and /b/ (according to Polish phonetic denomination). Table 2 presents the chosen phonemes. Table 3 presents the list of words used in the research.

**Table 2.** Investigated phonemes were combined into groups according to their place of articulation in Polish.

| Group of Consonants | Phonemes Included |
|---|---|
| Bilabial | /b/, /p/ |
| Alveolar | /t/, /d/, /n/ |
| Labiodental | /f/, /w/ |
| Palatal | /sz/, /ż/, /cz/, /dż/ |
| Palatal_PL * | /ś/, /ź/, /ć/, /dź/ |
| Dental | /s/, /z/, /c/, /dz/ |

* group of specific Polish regional palatal consonants which are not represented by the International Phonetic Alphabet.

**Table 3.** The list of words included in the Articulation Test for Children and Adolescents.

| Group of Consonants | Words Included in the Test with Their Word Position (Front, Middle, Final) |
|---|---|
| Bilabial * | **B**a**b**cia, **b**alon, dia**b**eł, ku**b**ek, **b**ęben, far**b**y<br>Pają**k**, **p**apuga, ła**p**a, ma**p**a, **p**tak, skle**p** |
| Alveolar * | /t/, /d/ *, /n/<br>**T**ory, **t**elefon, bu**t**y, kwia**t**y, samolo**t**, pło**t**<br>**D**ywan, **d**ym, bu**d**a, pu**d**ło, **d**eszcz, bie**d**ronka<br>**N**amiot, **n**uta, kola**n**o, koro**n**a, wago**n**, bana**n** |
| Labiodental * | /f/, /w/<br>**F**otel, **f**oka, ła**w**ka *, wa**f**le, mikro**f**on, le**w** *<br>**W**orek, **w**anna, owo**c**e, kro**w**a, **w**łosy, kra**w**at *<br>**In ławka /w/ is pronunced as voiceless /f/ because of /k/.**<br>**/w/ is always pronunced voiceless in the final position.** |
| Palatal | /sz/, /ż/, /cz/, /dż/<br>**Sz**alik, **sz**afa, wie**sz**ak, mu**sz**la, my**sz**, kapelu**sz**<br>**Rz**eka *, **ż**aba *, je**ż**e, o**rz**ech, g**rz**yby, ły**ż**wy<br>**Cz**ajnik, **cz**apka, tę**cz**a, ka**cz**ka, klu**cz**, mle**cz**<br>**Dż**ungla, **dż**em, **dż**okej, **dż**okejka, zje**żdż**alnia, **dżdż**ownica |
| Palatal_PL * | /ś/, /ź/, /ć/, /dź/<br>**Si**atka *, **ś**winia, hu**ś**tawka, wi**ś**nie, gę**ś**, mi**ś**<br>**Zi**eminiaki, **zi**ma, ba**zi**e, guzi**ki**, łazi**enka**, gałę**zi**e<br>**Ci**eń, **ci**astko, po**ci**ąg, bo**ci**an, paznokie**ć**, niedźwie**dź** *<br>**Dzi**eci, **dzi**upla, bu**dzi**k, łabę**dzi**e, **dź**wig, gwoź**dzi**e |
| Dental | /s/, /z/, /c/, /dz/<br>**S**anki, **s**owa, pa**s**ek, para**s**ol, no**s**, pie**s**<br>**Z**apałki, **z**amek, ko**z**a, wó**z**ek, **z**nak, liz**a**k<br>**C**ukierek, **c**ebula, ta**c**a, ple**c**ak, widele**c**, pale**c**<br>**dz**won, **dz**banek, ro**dz**ynki, kukury**dz**a, pę**dz**el, pienią**dz**e |

* Phonemes /b/, /w/, /d/, /ż/, /dż/ /ź/, /dź/, /z/, and /dz/ in their final position are voiceless. In Polish, soft consonants are marked with an accent above the letter when they appear at the end of words (i.e., koń) and when they appear before the letter denoting a consonant (i.e., ćma). We denote softness in the sounds of a silent row with the letter "i", when the sound "i" is syllabic (i.e., siatka, ciastko), and when it is in front of the vowels. The letters "rz" and "ż" mean the same phoneme. The spelling depends on the Polish orthography.

The study was conducted and recorded with a voice recorder by a qualified speech pathologist. During a basic intraoral examination, the assessment included: the type of occlusion (noted after an orthodontist or a maxillofacial surgeon diagnosis), teeth position (including incisors protrusion or retrusion), missing teeth, crowding, diastema, and other dental malposition; the shape of the tongue and its resting position; the tongue and lip frenulum length; the shape of dental arches and palate; breathing, swallowing, biting, and chewing patterns; and the pace of speech and its fluency. Furthermore, the

patients were asked to subjectively assess their hearing. They also answered a question about previous speech therapy in childhood (Yes/No). The motor assessment of the tongue and lips was based on the aforementioned adaptation of the Speech Organ Fitness Questionnaire [21], in which 20 tests were provided to assess both tongue and lip motor skills. The questionnaire has been commonly used among Polish speech pathologists. Each trial was rated on a scale from 0 to 3 points, where 0 indicated impossible movement and 3 indicated perfect movement. The next step was to conduct a phonetic analysis. Each phoneme was assessed in 3 different voice word positions, i.e., front, middle, and final. During the articulation examination, both auditory and visual analyses were used to determine the correct positioning of the speech organs, i.e., the appropriate position of the tongue, lips, and cheeks for the given phoneme defined as a norm in phonetics. For the visual analysis, some primary criteria were established: a 0- to 3-point scale, where 0 stands for an articulation movement done completely incorrectly in which both lips and tongue were not appropriately shaped, the tongue being in a different place than it should be (for example, it was placed in an interdental manner); 1 stands for better lips position, i.e., round lips for the rustling phonemes position, but still an incorrect positioning of the tongue, i.e., lateral or interdental; 2 stands for regular lips position but lateral and low tongue, i.e., rustling phonemes were pronounced at the height of the central incisors and not the palate; and 3 stands for perfect motor skills of lips and tongue. The pronunciation of a particular phoneme was in agreement with the articulation place and was taken as the main factor conditioning the assessment as a correct or an incorrect one. Moreover, for these three groups of tests, the individual scores were summed, and the pooled scores were evaluated as Total Lips Efficiency, Total Tongue Efficiency, and Total Visual Assessment.

In the presented study, the correct pronunciation was examined in phonemes in words, taking into account the different positions of phonemes in the word. The normative sound was the one in which no undesirable phonetic features were detected. The desired phonetic features are compatible with their articulation structure, with the description of the normative phoneme and its features, and they are compatible both acoustically and visually. In an incorrect pronunciation, undesirable phonetic features are those which, in the visual assessment, deviate from the articulation standard and the place of articulation and whose sound differs from the pattern established in phonetics. This division was based on studies by Konopska [20], who created her own classification of undesirable phonetic features. The basis of the nomenclature is the names of the articulators or the places of articulation, i.e., bilabiality, laterality; acoustic tone, i.e., dysdentalization; and undesirable additional articulation, i.e., passive or active labial activity (i.e., pronouncing /b/ or /p/ with central incisors placed on the lower lips instead of labial closure). The speech assessment included a visual and acoustic observation during the pronunciation of sounds in words and in the logotome (CV), i.e., /sa/, /so/, /se/.

The classification of the articulation place was adopted after T. Laine's [8] study protocol as follows: anterior variant, which means that the phoneme was pronounced too closely to the teeth; interdental variant, when the tongue was located precisely between the central incisors, without performing lateral movements; and lateral variant, which indicates a deviation of tongue movements to the right or left and posterior variant, indicating that phonemes were pronounced with the tongue mass withdrawn towards the oropharynx. A 5-grade scale of pronunciation was established based on similar paediatric research by Krasnodębska [21] (such a scale allowed for assessing if there were any undesirable phonetic features): 1 point meaning phoneme omission; 2 points meaning a completely deformed phoneme pronounced in an interdental manner; 3 points meaning that the phoneme was pronounced too anteriorly; 4 points indicating that a phoneme was pronounced with a lateral position; and 5 points meaning that the phoneme was pronounced correctly. The data were collected before the surgery during the orthodontic treatment and 3 months after the surgery was completed. All the participants in the research group had at least 3 sessions with the speech therapist. In the first session, the diagnosis was taken. During the second session, the patients were provided with basic speech therapy and

guidance along with some simple exercises on proper swallowing and breathing. During the second meeting, the speech therapy was continued, and the patients also received a set of exercises intended to help them recover after surgical intervention. Between the sessions with a therapist, patients were asked to train at least 20 min daily. The last meeting was the check-up, 3 months after the surgical procedures, and the second complete examination was conducted, maintaining the previous procedure.

Statistical analysis was performed in Statgraphics Centurion 18 (Statgraphics Technologies Inc., The Plains, VA, USA). Averages before and after surgery were compared by a paired *t*-test. Categorical variables were tested for independence by the chi-square test. One-way ANOVA or Kruskal–Wallis (as a lack of normal distribution was observed) tests were applied for a factor (categorical variable) influence evaluation onto quantitative variables. The relationship of two quantitative variables was investigated by simple regression. Linear regression was performed in which correlation coefficient (CC) calculations and the R-squared statistic ($R^2$) indicated how much the model fit to the variability in post-operational consonants. The possibility of grouping consonants of a different and higher order than that typically proposed by the International Phonetic Association was tested. Factor analysis was conducted for this purpose. Significant statistics were considered as eigenvalue > 1.0 and $p < 0.05$.

## 3. Results

All the patients had a history of speech therapy in childhood and still faced some articulation defects of varying severity: light, mild, or heavy. Most patients had some minor difficulties with lip and tongue movements. Tables 4 and 5 present the results of tongue and lip motor skills before and after the treatment.

Assessment of functional lip performance (Table 4) in the "Lip movement in-out" test did not improve significantly after the treatment but was better than in the reference group ($p < 0.01$); moreover, preoperational function was also better than the reference ($p < 0.01$). The exact relations were found for "Lips movement right-left," "Circle," "Wild smile, and "Fish mouth" tests, but with higher significance ($p < 0.001$). In "Kiss" and "Vibration" tests, the postoperational function was better than in the reference group ($p < 0.01$) but did not differ from the preoperational function (the preoperational function also did not differ from the reference). No differences between the three study groups were found in the tests "Lips shot" and "Lips up-down". An improved postsurgery function of the lips was noticed in the "Teeth lips catch" test ($p < 0.001$) as opposed to the preoperational examination, as well as in the case of the reference group. While considering "Total Lips Efficiency", it is essential to note the significant improvement in lip function after treatment, both compared to the pretreatment study and to the reference group ($p < 0.001$).

**Table 4.** Lip function in operated dysgnathic patients.

| Lips Function Test | Reference Average ± SD | Preoperational Pre-OP Average ± SD | Postoperational Post-OP Average ± SD | Statistical Significance Pre-OP vs. Post-OP |
|---|---|---|---|---|
| Lips Movement In-Out | 2.46 ± 0.69 | 2.65 ± 0.54 | 2.78 ± 0.42 * | n.s. |
| Lips movement Right-Left | 1.78 ± 0.89 | 2.27 ± 0.69 * | 2.51 ± 0.73 * | n.s. |
| Kiss | 2.30 ± 0.74 | 2.62 ± 0.83 * | 2.78 ± 0.48 * | n.s. |
| Lips Shot | 2.00 ± 1.11 | 1.95 ± 1.05 | 2.32 ± 0.85 | n.s. |
| Vibration | 2.22 ± 1.11 | 2.46 ± 0.96 | 2.78 ± 0.48 | n.s. |
| Circle | 2.11 ± 0.91 | 2.59 ± 0.60* | 2.65 ± 0.75 * | n.s. |
| Wide Smile | 2.19 ± 0.78 | 2.78 ± 0.53* | 2.84 ± 0.44 * | n.s. |
| Teeth Lips Catch | 1.46 ± 0.87 | 1.38 ± 0.79 | 2.24 ± 0.86 * | $p < 0.001$ |
| Fish Mouth | 2.19 ± 0.88 | 2.51 ± 0.65 * | 2.76 ± 0.49 * | n.s. |
| Lips Up-Down | 2.32 ± 0.88 | 2.24 ± 0.80 | 2.57 ± 0.65 | n.s. |
| Total Lips Efficiency | 21.53 ± 4.44 | 23.49 ± 2.80 * | 26.08 ± 4.52 * | $p < 0.001$ |

Abbreviations: n.s.—no significant difference; *—significantly better function compared to reference ($p < 0.05$).

**Table 5.** Lingual function in operated dysgnathic patients.

| Tongue Function Test | Reference Average ± SD | Preoperational Pre-OP Average ± SD | Postoperational Post-OP Average ± SD | Statistical Significance Pre-OP vs. Post-OP |
|---|---|---|---|---|
| Tongue Down | 2.46 ± 0.65 | 2.70 ± 0.52 | 2.73 ± 0.45 | n.s. |
| Tongue Up | 1.70 ± 1.13 | 2.24 ± 0.95 * | 2.35 ± 0.82 * | n.s. |
| Tongue Lips Corner | 2.73 ± 0.51 | 2.86 ± 0.42 | 2.89 ± 0.39 | n.s. |
| Wide Tongue In-Out | 2.41 ± 0.76 | 2.51 ± 0.61 | 2.73 ± 0.45 | n.s. |
| Spoon Tongue | 2.19 ± 0.94 | 1.95 ± 0.97 | 2.24 ± 0.72 | n.s. |
| Arched Tongue | 1.81 ± 1.05 | 1.43 ± 1.09 | 2.05 ± 0.81 | $p < 0.05$ |
| Tongue Back | 2.27 ± 0.65 | 2.76 ± 0.49 * | 2.73 ± 0.56 * | n.s. |
| Lips Lick | 2.43 ± 0.67 | 2.76 ± 0.43 * | 2.73 ± 0.45 * | n.s. |
| Tongue Sound | 2.14 ± 1.03 | 2.43 ± 0.80 | 2.51 ± 0.80 | n.s. |
| Vertically Horizontal Position (Rest Position) | 1.32 ± 1.13 | 1.59 ± 1.01 | 1.95 ± 0.91 * | n.s. |
| Total Tongue Efficiency | 21.46 ± 4.90 | 23.27 ± 3.30 * | 24.73 ± 3.20 * | n.s. |

Abbreviations: n.s.—no significant difference: *—significantly better function compared to reference ($p < 0.05$).

The evaluation of the function of the tongue (Table 5) in the "Lips lick" test did not improve significantly after the treatment; however, the average score was higher than in the reference group ($p < 0.01$), and simultaneously, the preoperational function was better than in the reference ($p < 0.01$). The exact relations were observed for "Arched Tongue" ($p < 0.05$), "Tongue Up", and "Tongue Back" tests and "Total Tongue Efficiency" with higher significance ($p < 0.001$). In the "Vertically Horizontal Position (Rest Position)" test, the postoperational function was better than in the reference group ($p < 0.05$) but did not differ from the preoperational function (the preoperational function was also not different from the reference). No differences between the three study groups were found in the tests: "Tongue Down", "Tongue Lips Corner", "Wide Tongue In-Out", "Spoon Tongue", and "Tongue Sound".

In the experimental group, a significant increase of pronunciation quality of bilabial, palatal (including Polish palatal (Palatal_PL)), and dental consonants was noticed after the treatment in comparison to the reference group ($p < 0.01$–$p < 0.001$). Alveolar consonants improved ($p < 0.001$) in the experimental group postoperationally to the reference (unlike in case of the preoperational group), and the reference group was significantly worse than in the preoperational series. As far as the labiodental consonants were evaluated, postoperational results were significantly better ($p < 0.001$) than both preoperational and reference groups (patients in the preoperational group pronounced these consonants significantly worse than in the case of the reference group). See Table 6.

**Table 6.** Articulation before and after surgery divided into consonant groups according to the International Phonetic Alphabet revised to 2015.

| Consonants | Reference Average ± SD | Pre-Operational Pre-OP Average ± SD | Post-Operational Post-OP Average ± SD | Statistical Significance Pre-OP vs. Post-OP |
|---|---|---|---|---|
| Bilabial | 4.59 ± 0.80 | 4.43 ± 0.83 | 4.95 ± 0.33 | $p < 0.01$ |
| Alveolar | 3.55 ± 0.85 | 4.36 ± 0.85 | 4.68 ± 0.54 | n.s. |
| Labiodental | 4.38 ± 1.00 | 4.08 ± 1.14 | 4.89 ± 0.31 | $p < 0.001$ |
| Palatal | 3.89 ± 0.94 | 4.12 ± 0.90 | 4.69 ± 0.48 | $p < 0.001$ |
| Palatal_PL * | 3.66 ± 1.03 | 4.22 ± 0.97 | 4.82 ± 0.50 | $p < 0.001$ |
| Dental | 2.72 ± 0.67 | 3.49 ± 1.23 | 4.55 ± 0.69 | $p < 0.001$ |

Abbreviations: n.s.—no significant difference; * group of specific regional palatal consonants which the International Phonetic Alphabet does not represent.

It has been noted that the pronunciation of bilabial, labiodental, palatal, palatal_pl, and dental consonants improved after treatment. We examined which of the study variables were associated with this improvement. The quality of the obtained pronunciation of bilabial consonants did not depend on the postoperational Total Visual Assessment (CC = 0.16, $R^2$ = 2.6%, $p$ = 0.35), Total Tongue Efficiency (CC = 0.07, $R^2$ = 0.5%, $p$ = 0.67), or Total Lips Efficiency (CC = 0.32, $R^2$ = 10%, $p$ = 0.055). Postoperational labiodental consonant quality did not depend on the Total Visual Assessment (CC = 0.07, $R^2$ = 0.6%, $p$ = 0.66), Total Tongue Efficiency (CC = 0.07, $R^2$ = 0.5%, $p$ = 0.69), or Total Lips Efficiency (CC = −0.13, $R^2$ = 1.8%, $p$ = 0.43). Next, the postoperational palatal consonants did not depend on the Total Tongue Efficiency (CC = −0.24, $R^2$ = 6%, $p$ = 0.15) or Total Lips Efficiency (CC = 0.03, $R^2$ = 0.1%, $p$ = 0.85) but did depend on the Total Visual Assessment in a relatively weak way (CC = 0.43, $R^2$ = 18%, $p$ < 0.05, Figure 1). As far as regional palatals (Palatal_PL) were concerned, the postoperational speech quality was not related to the Total Tongue Efficiency (CC = 0.06, $R^2$ = 0.4%, $p$ = 0.70) or Total Lips Efficiency (CC = −0.1, $R^2$ = 1%, $p$ = 0.54) but was moderately more strongly related to the better postoperational Total Visual Assessment (CC = 0.65, $R^2$ = 42%, $p$ < 0.05). Finally, dental consonant pronunciation was not related to Total Lips Efficiency (CC = −0.05, $R^2$ = 0.2%, $p$ = 0.78), but a relatively weak relationship was found to the Total Visual Assessment (CC = 0.49, $R^2$ = 24%, $p$ < 0.05) and to the Total Tongue Efficiency (CC = 0.44, $R^2$ = 11%, $p$ < 0.05). See Figure 1.

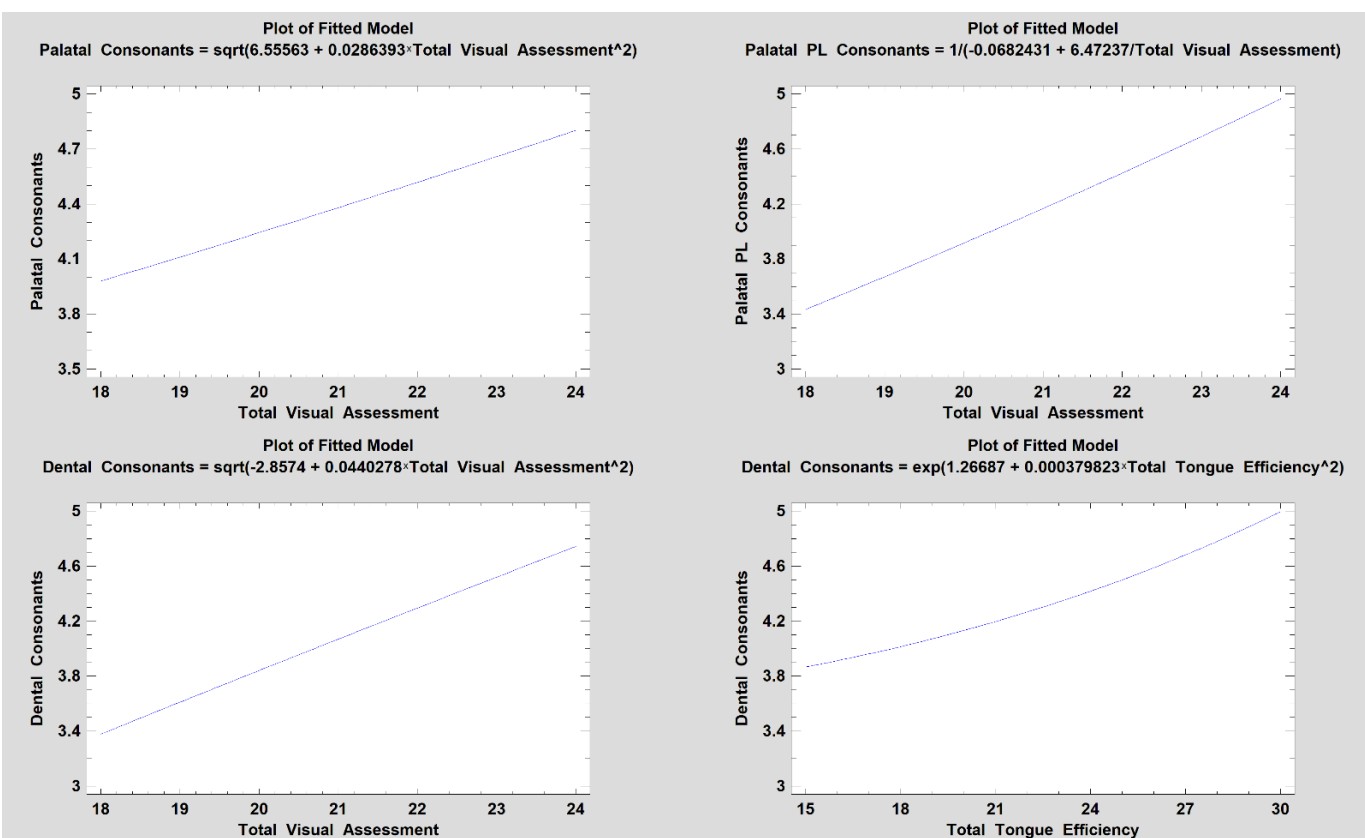

**Figure 1.** Final palatal, palatal_pl, and dental consonant articulation was related to the postoperational Total Visual Assessment ($p$ < 0.05). Such a relationship was also observed between dental consonants and total Tongue Efficiency assessed postoperatively ($p$ < 0.05).

When considering the effectiveness of speech therapy, it was noted that it was higher in patients with mandibular retrognathism than in patients with skeletal Class III in a group of palatal consonants (Kruskal–Wallis test, $p$ < 0.05). Furthermore, it was observed that specific dental components of a morphological defect were associated with better speech therapy outcomes (Figure 2). In uncomplicated open bite by any other dental

malposition and in patients with diastema, simple dental crowding, or possibly crowding with incisor retrusion, dental consonants were significantly better than after orthodontic-surgical treatment of open bites complicated by malposition of incisors or retrusion of incisors on the background of skeletal malposition Class II or III (Kruskal–Wallis test, $p < 0.05$).

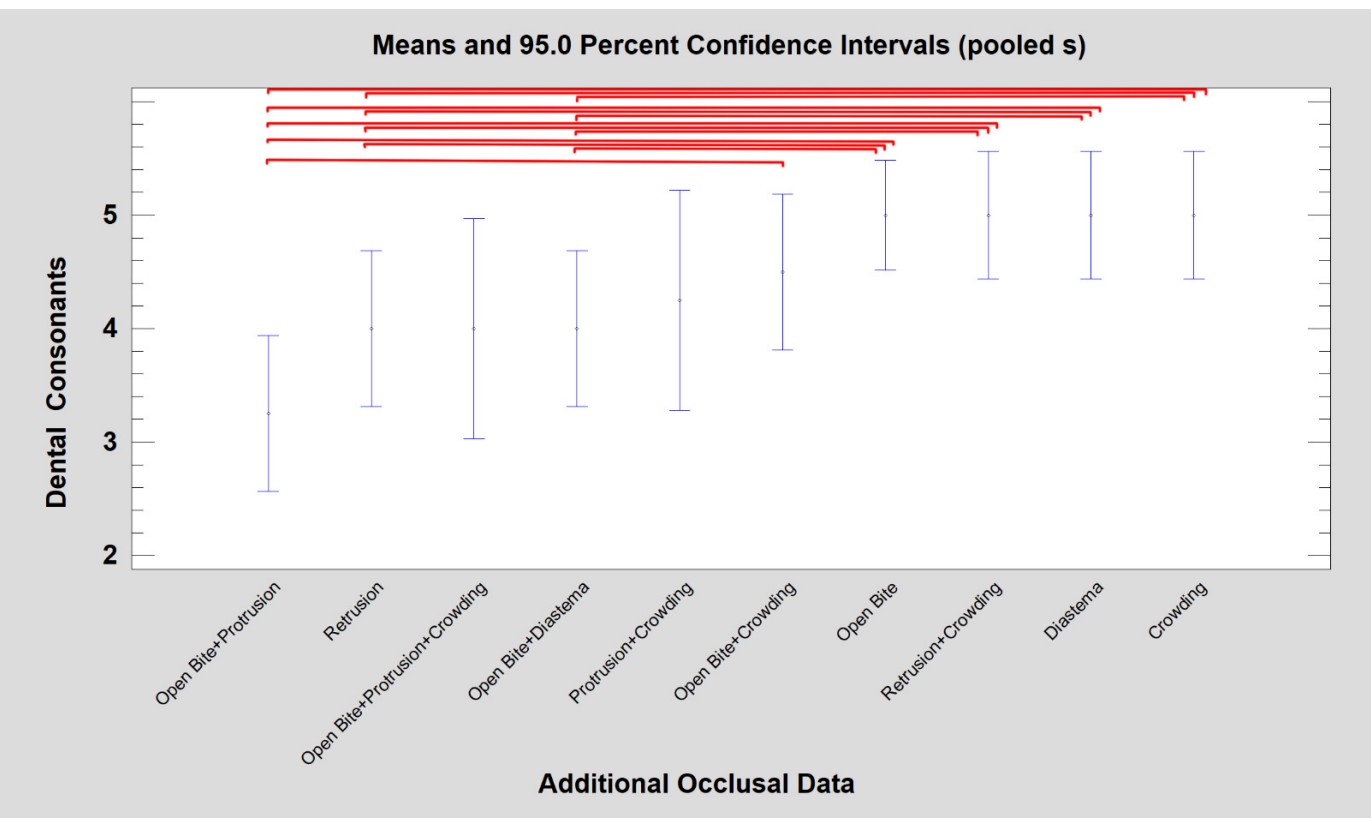

**Figure 2.** Final dental consonant pronunciation results in patients treated multimodally (higher scores, better speech). Quality of speech depends on specific dental positioning. These pairs show statistically significant differences at the 95.0% confidence level: Open Bite + Protrusion vs. Open Bite + Crowding, Open Bite + Protrusion vs. Open Bite, Open Bite + Protrusion vs. Retrusion + Crowding, Open Bite + Protrusion vs. Diastema, Open Bite + Protrusion vs. Crowding, Retrusion vs. Open Bite, Retrusion vs. Retrusion + Crowding, Retrusion vs. Diastema, Retrusion vs. Crowding, Open Bite + Diastema vs. Open Bite, Open Bite + Diastema vs. Retrusion + Crowding, Open Bite + Diastema vs. Diastema, and Open Bite + Diastema vs. Crowding ($p < 0.05$). Red brackets indicate statistically significant differences ($p < 0.05$).

The results of multimodal speech therapy for patients with facial skeletal deformities have already been analysed in this study. Now, with a large group of patients (all obtained presurgical results created 88 cases logopaedically tested), the effect of skeletal and dental malocclusion on consonant deformities will be evaluated below. Here, unlike the previous part of the results section, the consonants will not be combined into traditional groups according to the International Phonetic Association but according to the statistical similarity of the severity of misarticulation.

Unfortunately, it has been impossible to perform factor analysis on every single consonant by itself because there have been frequently similar pronunciation test results, which have induced linear dependencies between variables. However, it has been possible to perform factor analysis based on the consonant groups defined by the International Phonetic Association (IPA). Bilabials, alveolar, labiodental, palatal, Palatal_PL, and dental consonants have been combined into two factors with an eigenvalue higher than 1. Together they account for 57% of the variability in the original logopaedic data derived from the IPA groups. Since the principal components method was selected, the initial

communality estimates have been set to assume that all of the variability in the data is due to common factors. The equations which estimate the common factors after rotation have been performed. Rotation has been performed in order to simplify the explanation of the factors. The explanations of such calculated factors are: Skeletal Consonants and Soft Tissue Consonants. The rotated factors have the Equations

$$\text{Skeletal Consonants} = -0.216 \times \text{Bilabials} + 0.686015 \times \text{Alveolar} + 0.182536 \times \text{Labiodental} + 0.646587 \times \text{Palatal} + 0.76647 \times \text{Palatal\_PL} + 0.78603 \times \text{Dental} \quad (1)$$

$$\text{Soft Tissue Consonants} = 0.760932 \times \text{Bilabials} - 0.111715 \times \text{Alveolar} + 0.774208 \times \text{Labiodental} + 0.0366595 \times \text{Palatal} + 0.144866 \times \text{Palatal\_PL} - 0.111252 \times \text{Dental} \quad (2)$$

The values of the equations' variables have been standardized by subtracting their means and dividing by their standard deviations. This also shows the estimated communalities, which can be interpreted as estimating the proportion of the variability in each variable attributable to the extracted factors (Figure 3).

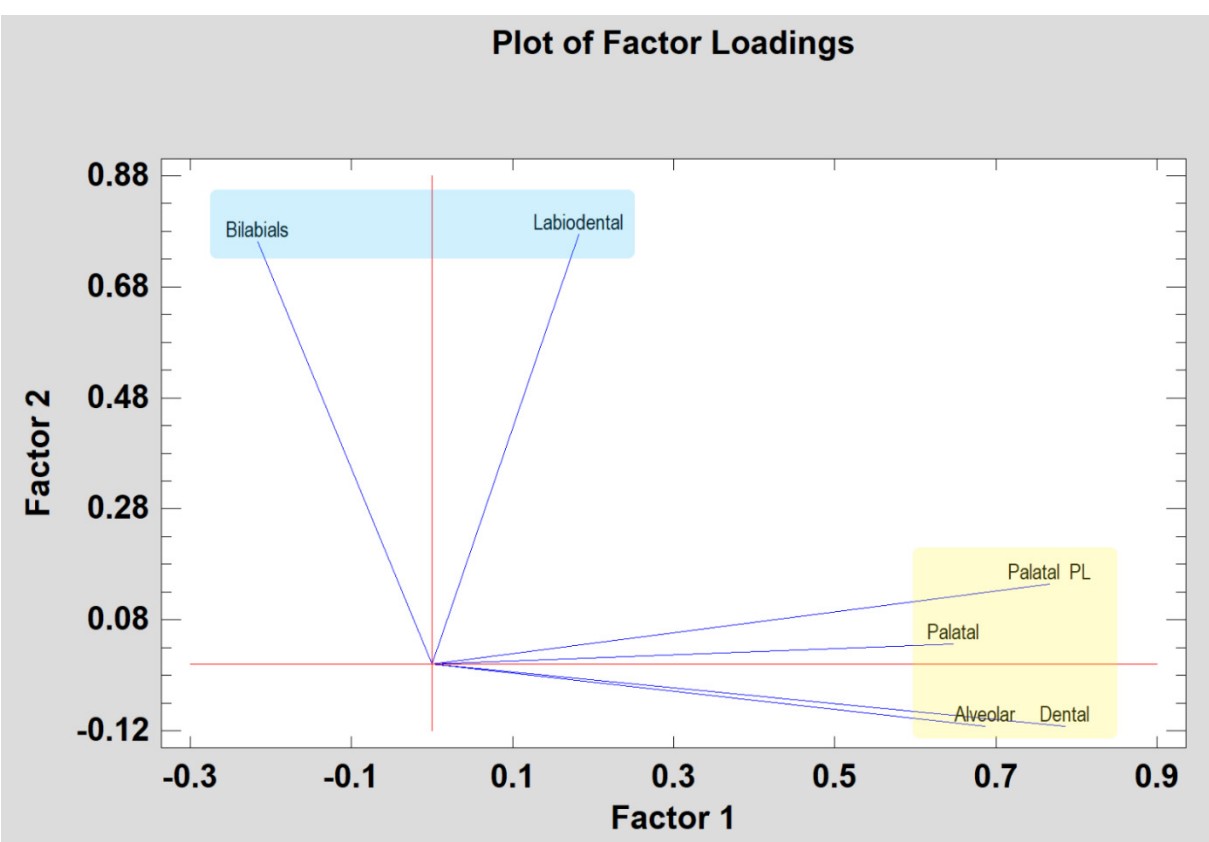

**Figure 3.** Components included in the two factors that achieved an eigenvalue higher than 1 (factor analysis). Factor 1 is called Skeletal Consonants (major components highlighted in yellow), and Factor 2 is called Soft Tissue Consonants (major components highlighted in blue). The data for 88 untreated patients.

Thus, the average value of the Skeletal Consonants factor calculated in presurgically evaluated patients was $10.55 \pm 2.14$, and the Soft Tissue Consonant factor was $6.71 \pm 1.17$. Skeletal and Soft Tissue Consonants are separate and not statistically related variables (CC = $-0.06$, $R^2 = 0.38\%$, $p = 0.5657$). Moreover, there was no statistically significant difference between the means of Skeletal Consonants ($p = 0.1827$) and Soft Tissue Consonants ($p = 0.7782$) from one level of gender to another.

The analysis of the variance provided information on the association of Skeletal and Soft Tissue Consonants with skeletal deformity (Figure 4). It appeared that the quality of pronunciation of Soft Tissue Consonants did not depend on the skeletal defect in the

craniofacial region ($p$ = 0.1507). In contrast, Class III maxillofacial deformity significantly worsened the pronunciation of Skeletal Consonants ($p < 0.01$). The worst results occurred in patients with mandibular prognathism (consecutively mandibular laterognathism). There was no association of dental malocclusion (variable: Additional Occlusal Data) with pronunciation quality for Skeletal Consonants ($p$ = 0.4102) or Soft Tissue Consonants ($p$ = 0.0533).

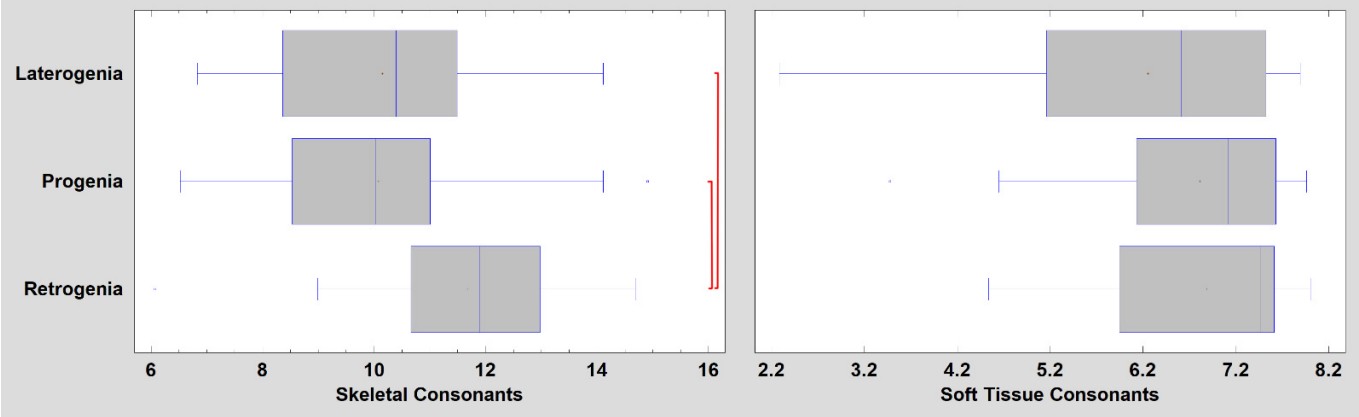

**Figure 4.** Effects of skeletal deformity on Skeletal Consonants and Soft Tissue Consonants. The pronunciation quality of Skeletal Consonants in patients affected by Skeletal Class III was significantly worse than in Class II patients. Red brackets indicate groups between which there is a significant difference in the quality of consonant articulation ($p < 0.01$).

Additionally, it was noted that in cases where a skeletal defect is detected, speech therapy conducted in childhood is not able to improve the articulation quality significantly, as these cases still presented the misarticulation of many phonemes. Furthermore, even patients treated in childhood were characterized by worse Skeletal Consonant pronunciation tests than adults (Figure 5, $p < 0.05$). This peculiar relationship was not observed in the Soft Tissue Consonants group ($p$ = 0.8381). Those results need to be carefully tested in future research, as it is still not clear if the faulty articulation depends on the compensatory mechanism, which was established by therapy in childhood, or if it is a mechanism connected with anatomical growth and development, which was not observed before.

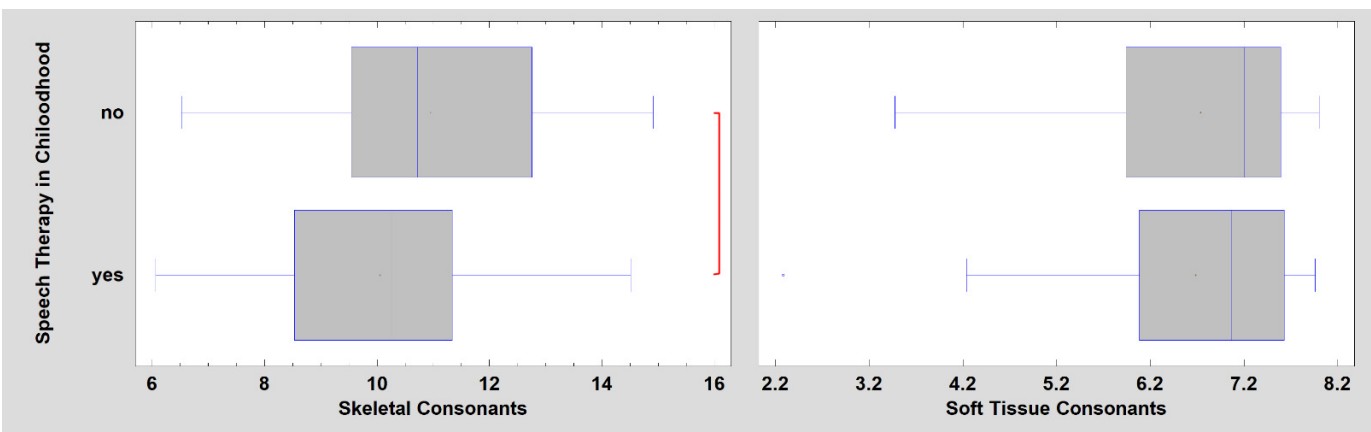

**Figure 5.** The relationship of speech therapy treatment in childhood to consonant pronunciation test results in adult patients affected by skeletal malformations. The red brackets indicate groups between which there is a significant difference in consonant articulation quality ($p < 0.05$).

The examination of sound production in these two groups of Skeletal and Soft Tissue Consonants also revealed a directly proportional relationship with three other aspects of patient examination: Total Lip Efficiency (CC = 0.37, $p < 0.001$ and CC = 0.26, $p < 0.05$, re-

spectively), Total Tongue Efficiency (CC = 0.24, *p* < 0.05 and CC = 0.32, *p* < 0.01, respectively), and Total Visual Assessment (CC = 0.57, *p* < 0.001 and CC = 0.37, *p* < 0.001, respectively).

## 4. Discussion

The associations between abnormalities in the teeth, mandible, and maxilla and articulatory speech disorders are mostly equivocal. The results presented in this study may highlight some areas which have been omitted or underestimated so far.

In 2007, Hassan et al. [23] published a review article on the effects of orthognathic surgery on speech, in which 18 longitudinal research studies conducted over the previous 50 years were evaluated. However, those reports were mainly limited and contradictory. The reason for the above resulted from the limited size of the examined groups, most of which lacked a reference group, and even if the reference was present, it was too small to compare appropriately. Many of them also did not have well-prepared statistical analyses. It was added that generally, every study differed by methodological assumptions. Some of them focused on the type of surgery, and some did not. The choice of testing tools and methods also differed. However, the most important and fundamental value of this article is that it proved how difficult it was to create a proper reference group. Considering reference groups, it must be noted that in Poland, children by the age of 3 are generally incorporated in speech therapy, which, in a standard way, lasts until they are 10–12 years old. This means that adolescents and adults with speech deficiency and malocclusion had finished their therapy by then and rarely attended speech therapy later on. The presented study focuses on the specific and multimodal approach to treat adult patients, taking into account their previous therapies.

To discuss the results, it is vital to point out that the Total Lip Efficiency has improved significantly. However, the Total Tongue Efficiency did not improve as immediately as it was expected to. As the articulation movements depend mainly on muscles, not bone, the recovery and repatterning of muscle functions take time. The patients were examined 3 months after the surgery. Considering that the lip functions are closely connected with the alignment of the maxilla and the mandible, the time of examination seemed to be perfect. The test of "teeth lips catch" proves easy to perform when the occlusion becomes normal. Moreover, there is no or weak muscle tension in the orofacial region, and some flexibility in the anterior-posterior movements of the mandible is visible. As far as the tongue functions are concerned, the evaluation time should be longer, as the recovery and remodelling process is prolonged and needs the assistance of a speech therapist. Perhaps the change in the skeletal determinants of tongue position and function is much more altered by surgery than the lips muscle position itself. The framing of the tongue by the mandibular body, the tension of the ligaments and muscles spanning between the mandibular arm and the base of the skull, and the tension of the suprahyoid muscles are greatly influenced by mandibular displacement during orthognathic surgery [15,24]. This may be seen in the "Arched Tongue" test in the preoperational group in comparison to the preoperational and reference groups. This test is very difficult to patients prior to the surgery, especially due to the high palate. After the surgery, the shape and height of the palate changes, and it results in a better performance afterwards. There is no good literature that could be discussed in comparison with these findings. Thus, in previous research on tongue frenotomy effectiveness [19], it was highlighted that the tongue with a short frenulum impairs normal functions such as swallowing and resting position and may even affect articulation. One of the possible answers for why the tongue performance does not improve may be that some participants were asked to consider frenotomy before the orthognathic surgery, and they did not accept the treatment. Similar thoughts about atypical functions of the tongue were presented by Jean-Marc Foletti [25], who researched atypical swallowing and possible relapse in patients undergoing combined orthodontic-orthognathic treatment of an open bite. The stability of surgical treatment is multifactorial. The facial muscle balance mainly depends on functions that could cause instability and lead to relapse. Foletti suggests that breathing, swallowing, and chewing disorders may be

the source. This research did not prove this; however, it may be seen that the abnormal function of the tongue has its impact on speech disorders. L. Vallino [13,14] pointed out that after orthognathic surgery, some patients, especially with closed bite, seemed to have a difficult time in adjusting to a newly created oral environment, which would promote improved articulatory positioning. She also added that in most of these cases, subjects tried to maintain an abnormal tongue position along with its functions, such as swallowing. The present study also confirms the statement by Vallino that the tongue can make adjustments, but it takes time.

Pronunciation and phonetics are an important part of this study. The experimental group achieved better pronunciation quality in almost every phoneme tested. Patients after combined treatment had a quicker recovery to normal speech. This research is parallel to that of Laine [8–12] and Vallino [13,14], who pointed out that some sounds are more sensitive to skeletal deformities than others. This study shows that only alveolar consonants did not improve significantly, and this may be related to the tongue functions which need more time to gain a new position and mobility. The results proved that the dental sounds /s/, /z/, /c/, and /dz/ are by far better pronounced by postoperational patients than by preoperational ones. This is because the maxilla and mandible are in normal occlusal relations. There are no undesirable phonetic features such as improper dentalization, laterality, additional areas of constriction, or active or passive labio-dental activity.

Moreover, those patients also do not have crowding, diastemas, protrusion, or retrusion in the arches, making the acoustic layer of pronunciation clear and comprehensive. Laine and Vallino stated that /s/ and /z/ are particularly prone to misarticulations, proven here. In Vallino's work, 30 out of 34 patients eliminated the errors or reduced them in number. She also noted that the most remarkable change was observed after 3 months postoperatively, similar to the present paper.

Furthermore, Vallino also pointed out that bilabial consonant /p,b/ errors were entirely eliminated after 3 months, and tip-alveolar /t,d,n/ errors needed more time and were eliminated after 6 months. The findings presented in Laine's, Vallino's, and our own results support the statement that orthognathic surgery used to correct skeletal and dental deformities leads to articulation improvement. Moreover, essential for open bite, Vallino highlighted that 9 out of 10 subjects had undergone previous unsuccessful speech therapy for improvement of dental consonants. These findings of Vallino's support the results presented here that the idea of speech therapy is not effective as the only method of treatment for speech disorders in cases of malocclusion in adults. Only a combined orthodontic–orthognathic treatment of malocclusion with speech therapy assistance may eliminate non-normative articulation. Another justification for the multimodal approach was provided by Mumim [26], who stated that the presence of an open bite could result in the tongue protruding into the space created between the upper and lower dental arches. It can be interdental or even lateral during articulating dental and palatal sounds. He also found out that if placed more forward, the tongue forces the mandible to be more depressed during articulation of the non-normative /s/, and if the open bite is large enough to prevent the lips from touching, then bilabial sounds are impossible to produce. This leaves a space for introducing some compensations, which are visible, such as placing the lower lip against the upper teeth, which is an incorrect placement for articulation (undesirable phonetic feature—wrong articulation place and manner). The present study proved that the labiodental consonants /f, v/ are better pronounced instantly. This is because the mandible and maxilla are in their proper positions, and, by this, the teeth also gain good occlusion immediately after surgery. When the occlusion such as in Class III is present, the teeth are prevented from contact with the lips and distort the sound. This example promotes the usage of visual and acoustical analyses simultaneously to assess if the phoneme has all desirable phonetic features. An experienced speech therapist should not base the assessment only on acoustics, as patients with malocclusion often use compensatory mechanisms (i.e., retrogenia patients often move the mandible anteriorly to pronounce /sz/ and by this, produce a good sound; however, the place of articulation,

the area of dentalization, and the constriction are improper, termed here an undesirable phonetic feature).

In Laine's works, the misarticulation of /s/ was clearly connected with mandibular overjet, and these results are parallel to the findings presented here. The results mentioned above proved that some additional occlusal factors such as crowding, a diastema, and retrusion might result in pronunciation disorders. Her work highlighted that in subjects with the absence of permanent teeth, the articulation of /r/, /l/, /n/, and /d/ was affected. This makes our research even more important as it proves that some structural deviations near the area of constriction (e.g., lower or upper teeth retrusion) change the airflow and influence the acoustic layer of the phoneme. In subjects with correct occlusion, a normative Polish /s/ would need a slightly protruded mandible, the edge-to-edge incisor relation, and a slightly elevated hyoid bone. Such adjustment would be impossible for subjects with Class III or those with open bite. As Laine stated in Class III, a more elevated hyoid bone and a simultaneous contraction of the hyoglossus muscle flattening prevent the tongue from forming a central groove, which is essential for the production of /s/. Moreover, she stated that alveolar consonants also require a large oral cavity anterior to the lingual constriction [12], which may not be possible in the case of the small maxilla [24].

Given the association of the pronunciation quality of the Skeletal Consonants with Class III skeletal deformities, the inclusion of such a defined group of consonants (Equation (1)) for the logopaedic evaluation of adults requiring speech therapy is worth considering. This should not be surprising since functional compensations of the tongue and lips cannot compensate for the severe structural abnormalities of the maxilla and mandible in Class III patients [3]. Moreover, patients treated in childhood are characterized by worse results of the Skeletal Consonants pronunciation tests in adulthood. Perhaps this is related to the lack of effective and nonstandardized speech therapy techniques used for children with facial skeletal deformities. It is also because, in children, speech therapists want to evoke phonemes that are not present or deformed, and this is often done through the usage of compensations in the orofacial region. By this, in some cases, malocclusion may develop faster, and more undesirable phonetic features may be observed. Undoubtedly, treating children with Class III skeletal deformities is always a challenge [13].

It seems uniform for different languages that the relationship between complicated open bite, mandibular prognathism, maxillary retrognathism, and faulty production of dental consonants exists. Moreover, the other risk factors for non-normative consonant articulation are dental anomalies such as protrusion, retrusion, or crowding, as they affect the acoustic layer, not the articulation place itself.

## 5. Conclusions

One of the most critical risk factors for misarticulations, as far as Polish consonant phonemes are concerned, is occlusal anomalies, which result in the changes in the tongue and lip positions and their functions. Moreover, occlusal deformations affect primary functions, such as breathing, swallowing, and chewing, which are inseparable elements for normative articulation. The interdisciplinarity of speech pathology and its effectiveness in the treatment of the adult population require the cooperation of a speech pathologist, a maxillofacial surgeon, an orthodontist, and even a physiotherapist to help patients improve their articulation and implement elements of myofunctional therapy.

Both Class II and Class III skeletal defects were found to respond equally well to speech therapy in adults (as a part of multimodal treatment). However, it was noted that in the treatment of a skeletal malocclusion, certain dental conditions allow for more effective speech therapy (i.e., dental crowding, presence of diastema, combinations of incisor retrusion with crowding, or even an open bite not accompanied by other dental determinants).

Only by teamwork combined with a patient's will can the goal of perfect articulation be achieved. Orthodontic preparation with surgical correction of malocclusion is often the only approach that helps to improve articulation, which is why such research needs to be conducted. The authors of the paper see two weaknesses in the study conducted. The first

is the inability to create a reference group of adult patients with speech impairment and without maxillofacial deformity. Additionally, the second is the lack of the possibility of studying patients from other language families.

**Author Contributions:** Conceptualization, A.L. and M.K.; methodology, A.L.; software, M.K.; validation, A.L. and M.K.; formal analysis, M.K.; investigation, A.L.; resources, A.L.; data curation, A.L. and M.K.; writing—original draft preparation, A.L.; writing—review and editing, A.L. and M.K.; visualisation, M.K.; supervision, M.K.; funding acquisition, M.K. All authors have read and agreed to the published version of the manuscript.

**Funding:** This research was funded by the Medical University of Lodz grant number: 503-1-138-01-503-51-001-17, 503-1-138-01-503-51-001-18 and 503-1-138-01-503-51-001-19-00.

**Institutional Review Board Statement:** The study was conducted according to the guidelines of the Declaration of Helsinki and approved by the Institutional Ethics Committee of the Medical University of Lodz (protocol code RNN/73/19/KB approval date: 12 February 2019).

**Informed Consent Statement:** Informed consent was obtained from all subjects involved in the study.

**Data Availability Statement:** Not applicable.

**Conflicts of Interest:** The authors declare no conflict of interest.

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
