# Peer review of "The Logopedic Evaluation of Adult Patients after Orthognathic Surgery"

_applsci, doi:10.3390/app11125732_

Round 1
Reviewer 1 Report
This article was well re-written and now achieved the level of publication in Applied Sciences.
Author Response
Thank you so much for your review.
We are happy to read that our work has been well reorganised.

Reviewer 2 Report
Overview of this paper:
This paper is described in comparisons of evaluation with orthodontic and orthogenetic treatments mainly, and then is evaluated by statistical analysis. By the treats, the treatments are improved on few functions such as palatal, alveolar, fricatives and so on. Authors concluded that surgical correction of malocclusion can expect to improve functions which utters as vocalized phonemes and syllables.
Comments:
For experiment, the authors collect the sample data such as clinical data including pre or post-surgery and treatments with each function and grouped functions at phonemes. This setup is so hard working with many times for collecting samples. However, the setup for trimming phoneme sections and its detail is unclear for understanding procedures. For example, speech sound such as phoneme is difficult to collect a pronounced sound as only phoneme. It is difficult to utter a phoneme only, because most of all language is composed from phoneme with vowels. By this reason, the pronounce should be recorded as CV (Consonant and Vowel) and/or CVC. So it is difficult to recorded a sound such phonemes on Table 2. In addition, the recoded sound has acoustic conjunction between before phoneme and after phoneme. Maybe you expect to record /b/, however the pronounced sound affects by dependent between consonant and vowel. By this reason, you should show the procedure which is eject to dependency of phonemes, or record the sound which is collected by independent pronunciation. If you trimmed the affection of dependency between before and after phoneme, please describe the detail and its reason which you collect the speech data.
Each experiments evaluated confirmation with significance which conditioned as <0.05. This is only confirmed that there is relationship between subjects or not. Although each result confirmed a significant difference between the two subjects, I thought that it did not prove that the vocalization and voice were improved. Can you hear and understand that the pronunciation was improved by listening to the sounds uttered from each phoneme?
This study employ 5 grade scale of pronunciation for all phoneme. However, I did not understand to choose this major for evaluation of significant on phoneme.
For evaluating the phoneme sound, most important is hearing expression for human subjects which the hearing ability has pathway between canal on ear to auditory cortex on brain. If you do not employ the subject experiment, I suggest you to evaluate pronunciations of phoneme with metrics modeled auditory nerve such as sound evaluation. There are some things that have not been fully considered and inferred just by showing the results of how to organize the figures. In Fig. 4, it is affected by the upper and lower jaws, isn't it? How does the phoneme that is uttered change as a tone control mechanism? It can be considered how the phonetic changes in physical changes and vocalizations improve hearing when the distance between the tongue and gums changes. The experimental objectives and recorded data are good, but the evaluation scale and considerations should be reconsidered.
I am looking forward to change a good scientific paper, thank you.
Author Response
Dear Reviewer
Thank you for your many valuable comments and tips. We made some changes to the article.
Firstly, we have added an explanation about the quality of the tested sounds was about and a list of words used in the study.
The speech pathology examination concerned not only the sound quality of the examined sounds, but also the arrangement (placement) of the organs. We did it based on the research of Dr. Konopska, who was the first in Poland to introduce the concept of undesirable phonetic features, and based on the phonetic norms of the Polish language, which indicate the above-mentioned features. For example, laterality, non-center formation of articulation gaps with the middle position of the tongue. With regard to, dentalized phonemes, this is the deviation of whole or only the front of the tongue from the centerline and improper airflow. In the case of dentalized phonemes, even a small non-occlusal space or the reverse overlap of the incisors are a very high risk factor in the occurrence of faulty phoneme implementation.
In the case of retrogenia, morphological changes, unless they are compensated by the forward movement of the mandible, reduce the articulation space and may affect the tongue movements. The consequence of retrogenia and the forward position of the lower arch when speaking is the exclusion of the inner surface of the upper incisors and part of the palate from the articulation.
Thus, the abnormal acoustics of the sound in such anatomical conditions is rather conditioned by the backward displacement of the main articulation site.
We agree that the degree of difficulty in speech pathology assessment increases when some disorders, e.g. horizontal bite, are accompanied by others.
The conducted research allowed to establish that malocclusion is a high risk factor for the occurrence of incorrect articulation.
Knowledge on this subject may have practical application in speech therapy, diagnosing and improving the pronunciation of people with a malocclusion.
Secondly, we add some interpretation of used tools.
The use of a 5-point scale makes it easier to identify an incorrect phonetic feature, because the scale determines the position of the articulation organs.
We also added some statements on using acoustic and visual assessments together. In Polish and world speech therapy, articulation assessment is more and more often based on both auditory and visual assessment. By combining these two methods of observation, it is possible to more accurately assess the occurrence of undesirable phonetic features and disorders of primary functions, such as e.g. swallowing. As we mentioned earlier, people with a malocclusion often use various compensation mechanisms that give us the impression that the sound is correct. However, taking into account the phonetic and linguistic aspects, one should also take into account the place and manner of producing a sound.One more important thing that should be emphasized is the fact that in our study we did not study the effect of operations on voice quality, but only on the articulation of individual phonemes in words and syllables.We are very grateful for the work you have done to assess our paper.
Yours faithfully,
Anna Lichnowska
Marcin Kozakiewicz
Reviewer 3 Report
The manuscript is interesting even if some aspects particularly the ones related to the presentation should be improved. I indicate some of the critical aspects and suggestions for the authors.
- The last part of the study relative to the assessment of the data of 88 patients (line 100, results from line 298) is not very clear and I suggest to the authors to redefine better the connection with the initial part. In particular, the n.88 patients are just mentioned at line 100 and the part of the study is not well defined in Methods in terms of relevance in relation to the first part (n.37 patients). Inclusions/exclusion criteria, age, medium age, sex, etc. are not indicated for the group of n.88 patients. This last aspect affects the presentation of the result at lines 349-363 since it is not clear how the aspects related to childhood were assessed. Furthermore, this part of the study as well as the reference group should be mentioned in the section “Abstract".
- Lines 377-378: The sentence should be revised since even if the manuscript has evidenced the difficulty related to the determination of the reference group, this aspect is itself also a limitation for the performed study as the same authors stated at the end of the discussion (line 498). Thus, the authors should revise the sentence by indicating better already at lines 377-378 that the reference group is a limitation of the study.
- The section “Discussion” is long-winded in some paragraphs, as at lines 437-460. The reported evidence from the literature should be discussed more briefly.
- Limitations of the study should be also reported in the section “Conclusions”.
- Line 156: Please, add a reference for the “Speech Organ Fitness Questionnaire”.
- Line 267: Please, check the number of the mentioned figure (Figure 7).
Author Response
Dear Reviewer,
Thank you very much for pointing out issues for improving our work.
We corrected the number of patients indicated in point 1. Responding to the indicated criteria, we would like to inform you that gender was not the selection criterion. '
In response to point 2, the indicated aspect that was included in the conclusions.
We are aware that the length of the discussion may be difficult for our reader, but the complexity of the research forces it to be properly described and related to world publications.
We also made some other corrections: we added a list of research words, we indicated precisely the methodology of the research and the aspects that we paid special attention to, e.g. undesirable phonetic features and their explanations.
Moreover, we corrected the language and numbering of tables, figures and bibliographies.
We hope that the quality of our study and its description meet Applied Science publication standards.
Yours faithfully,
Anna Lichnowska
Marcin Kozakiewicz

Round 2
Reviewer 2 Report
Dear authors
I read your claim and then understood why you choose the 5 point majors. However, this outcome is limited whether your research field at on your country. This transaction is published for all over the world, and should acceptable procedures and process. This major of limitation is subjective and not statistical whether different phoneme and syllables. According to your response for my reviewer note. Your paper is claimed that the relationship between treatment and without treatment for malocclusions and “The forward movement of the mandible, reduce the articulation space and may affect the tongue movements”. For more precisely measurement, why you do not measure the effectiveness with cephalogram and other imagines? The information of 5-point major is compressed as result of hearing experiments.
If you can be accepting to change the expressions for “Multimodal and logopedic” into meaning for ”treatment evaluation for malocclusion with differences by 5 point mean (metrics)”.
The followings are conditions for paper suggestions to this transaction.
1. Change your research claim into expression “treatment evaluation for malocclusion with differences by 5 point mean (metrics)”. I welcome to accept other expressions when the limitation is not changed at meaning.
2. The related expression for adjustment of the revision should revise at whole your paper.
At next revision, I will decide to suggest your paper for accepting or not.
I would like to wait your good answers with the revisions, thank you.
Author Response
Dear Reviewer,
In the beginning, we would like to say that this is neither orthodontic nor phoniatric work, nor the scope of oral and maxillofacial surgery, nor even dentistry. It is simply a speech therapy publication. Of course, its background is orthognathic, but that is just a feature of the adult patient population studied. It is also an essential feature of this work because there is very little speech therapy research on adult patients and they mainly concern neurological disorders, neurodegenerative diseases, stuttering or disorders related to the rehabilitation after cancer. See below:
- Effects of Speech Therapy in Hospitalised Patients with Post-Stroke Dysphagia: A Systematic
Review of Observational Studies DOI: 10.20344/amp.9183
- Kelsall D, Lupo J, Biever A. Longitudinal outcomes of cochlear implantation and bimodal hearing in a large group of adults: A multicenter clinical study. Am J Otolaryngol. 2021 JanFeb;42(1):102773. doi: 10.1016/j.amjoto.2020.102773. Epub 2020 Oct 22. PMID: 33161258.
- Raglio A, Oasi O, Gianotti M, Rossi A, Goulene K, Stramba-Badiale M. Improvement of spontaneous language in stroke patients with chronic aphasia treated with music therapy: a randomised controlled trial. Int J Neurosci. 2016;126(3):235-42. doi:
10.3109/00207454.2015.1010647. Epub 2015 Jun 10. PMID: 26000622.
- Liu SY, Yu G, Lee LA, Liu TC, Tsou YT, Lai TJ, Wu CM. Audiovisual speech perception at various presentation levels in Mandarin-speaking adults with cochlear implants. PLoS One. 2014 Sep 15;9(9):e107252. doi: 10.1371/journal.pone.0107252. PMID: 25222104; PMCID: PMC4164527.
- de Bruijn MJ, ten Bosch L, Kuik DJ, Quené H, Langendijk JA, Leemans CR, Verdonck-de Leeuw IM. Objective acoustic-phonetic speech analysis in patients treated for oral or oropharyngeal cancer. Folia Phoniatr Logop. 2009;61(3):180-7. doi: 10.1159/000219953. Epub 2009 Jul 1. PMID: 19571552.
Unfortunately, speech therapy is primarily a speciality dealing with children's therapy [there are 32571 records found in PubMed as "speech therapy children" are used as keywords]. Moreover, adults are very poorly examined patients. Our paper tries to shed some light on this unknown and vital aspect of speech therapy. In our work, the applicable standards of speech therapy diagnosis are applied according to the International Phonetic Alphabet. The sounds selected for our inventory are indicated in world publications as the most disturbed:
1 Tellervo Laine, Malocclusion traits and articulatory components of speech, European Journal of
Orthodontics, Volume 14, Issue 4, August 1992, Pages 302–
309, https://doi.org/10.1093/ejo/14.4.302
- Tellervo Laine, Associations between articulatory disorders in speech and occlusal anomalies, European Journal of Orthodontics, Volume 9, Issue 2, May 1987, Pages 144– 150, https://doi.org/10.1093/ejo/9.2.144
- Tellervo Laine, Articulatory disorders in speech as related to size of the alveolar arches, European Journal of Orthodontics, Volume 8, Issue 3, August 1986, Pages 192–
197, https://doi.org/10.1093/ejo/8.3.192
- Comparative study of normal defective articulation of /s/ as related to malocclusion and deglutition J D SUBTELNY, J C MESTRE, J D SUBTELNY DOI: 1044/jshd.2903.269.
During the last 60 years, only 18 such research studies were created [DOI: 10.1016 / j.joms.2007.05.018] and all of them used auditory assessment referring to the normative description of the sound (the same as in our study). It was considered whether the sound spoken by the patient is correct or not [i.e. the result was two-degree = dichotomic = qualitative], without adding information about the occurrence of undesirable phonetic features such as improper dentalization, palatalisation or additional articulations. Our work in its 5-point scale [quasiquantitative evaluation is much better than qualitative one] determines not only whether a given sound is disturbed, but also gives a specific feature of the disorder {or the severity of the disorder?}. This is a novel approach to the subject, but we continued to study those sounds that have been described as typical over the past 60 years. Additionally, in the works of some researchers, e.g. Turvey, 1976 [doi.10.1016 / s0301-0503 (76) 80014-8], there are scores for the functioning of the lips, tongue, swallowing and lisp (0-3). Most of the tests examining articulation are designed for one selected language, the main criteria is the relationship of the spoken sound with the norm adopted from the International Phonetic
Alphabet [https://www.internationalphoneticassociation.org/sites/default/files/IPA_Kiel _2015.pdf]. Our division of sounds is consistent with the cited classification, and however, due to the language belonging to the group of Slavic languages, there are sounds called regional voicing (Palatal_PL). The 5-point scale determines which of the articulators has a limited or incorrect function and defines the type of pronunciation related to the current international classifications.
Additionally, our classification was based on the Finnish protocol by prof. Tellevro Laine, who conducted the most extensive and most important research so far, did not point to undesirable phonetic features but to the type of tongue positioning and identified anterior, posterior, lateral and unclassified language abnormalities. only 4-degree of freedom)
The variants presented by Laine concerned errors in the arrangement of organs in relation to the correct place of articulation (just like our scale, here); she also divided the pronunciation errors into 2 groups: grade 1 - the sounds were still understandable, grade 2 - severe deformations of the sounds disturbing the sound of a word or syllable [Please notice, that it is still only 2-degree i.e. less accurate than our 5-degree evaluation].
We would like to refer to the issue of an international group of readers versus patient research in the field of Polish. This is a common problem for speech therapists. Works from many language groups and countries are created and published in English:
- Hu W, Zhou Y, Fu M. [Effect of skeletal Class III malocclusion on speech articulation]. Zhonghua Kou Qiang Yi Xue Za Zhi. 1997 Nov;32(6):344-6. Chinese. PMID: 11189306.
- Wakumoto M, Isaacson KG, Friel S, Suzuki N, Gibbon F, Nixon F, Hardcastle WJ, Michi K.
Preliminary study of articulatory reorganisation of fricative consonants following osteotomy. Folia Phoniatr Logop. 1996;48(6):275-89. doi: 10.1159/000266422. PMID: 8958664.
- Khinda V, Grewal N. Relationship of tongue-thrust swallowing and anterior open bite with articulation disorders: a clinical study. J Indian Soc Pedod Prev Dent. 1999 Jun;17(2):33-9. PMID: 10863488.
- Vallino LD. Speech, velopharyngeal function, and hearing before and after orthognathic surgery. J Oral Maxillofac Surg. 1990 Dec;48(12):1274-81; discussion 1281-2. doi: 10.1016/02782391(90)90481-g. PMID: 2231145.
- Pahkala, T. Laine, M. Närhi, U-M. Ettala-Ylitalo, Relationship between craniomandibular dysfunction and pattern of speech sound production in a series of first-graders, European Journal
of Orthodontics, Volume 13, Issue 5, October 1991, Pages 378–
385, https://doi.org/10.1093/ejo/13.5.378
- Qvarnström, M. J., Laine, M. T., & Jaroma, S. M. (1991). Place of articulation in articulatory speech disorders of different sounds in a group of Finnish first-graders. Folia Phoniatrica, 43(4), 161– 170. https://doi.org/10.1159/000266120
- Tellervo Laine, Arja-Liisa Linnasalo, Marjatta Jaroma,
Articulatory disorders in speech among Finnish-speaking students according to age, sex, and speech therapy, Journal of Communication Disorders, Volume 20, Issue 4,
- Qyarnström M, J, Jaroma S, M, Laine M, T: Changes in the Peripheral Speech Mechanism of Children from the Age of 7 to 10 Years. Folia Phoniatr Logop 1994;46:193-202. doi:
10.1159/000266313
- McLeod S, Verdon S. A review of 30 speech assessments in 19 languages other than English. Am J Speech Lang Pathol. 2014 Nov;23(4):708-23. doi: 10.1044/2014_AJSLP-13-0066. PMID: 24700105.
- Turvey TA, Journot V, Epker BN. Correction of anterior open bite deformity: a study of tongue function, speech changes, and stability. J Maxillofac Surg. 1976 Jun;4(2):93-101. doi:
10.1016/s0301-0503(76)80014-8. PMID: 1065713.
- Bianchini EM, de Andrade CR. A model of mandibular movements during speech: normative pilot study for the Brazilian Portuguese language. 2006 Jul;24(3):197-206. doi:
10.1179/crn.2006.032. PMID: 16933461.
In the last 60 years, only 18 publications have reported the results of evaluating speech disorders in patients with skeletal malocclusion. Our paper is the nineteenth study. Below is a table 1 summarising the world-wide literature on this topic.
Kindly note that of the 18 articles, 14 do not have a reference group (I have highlighted this in red), of these remaining 4 studies, the reference groups are only 5-9 cases each (source: Hassan, T., Naini, F. B., & Gill, D. S. The Effects of Orthognathic Surgery on Speech: A Review. Journal of Oral and Maxillofacial Surgery, 2007, 65, 12, 2536–2543. doi:10.1016/j.joms.2007.05.018). It is also worth noting that half of this small pool where there is a reference group does not have the patients' age. Furthermore, in one of the two studies, the age is below 18 years, which means they are not adults. This is the state of research present in the literature. So there is only one study from the last 50 years in which there is a reference group made up of adult patients. Unfortunately, only nine adults. This shows how difficult (almost impossible) it is to create a reference group for adult studies. Against this background, I would like to emphasise that the adult reference group has a very large size in our study. It is the largest adult referral group known in the literature. The largest reference group known in the literature was included in our study.
To summarising, speech therapy is usually qualitatively assessed at two levels. Our study is unique because it:
- assesses up to 5-degree i.e. quasi-quantitative
- has a very large group of adults [one of the largest world-wide]
- includes adult study
Yours faithfully,
Anna Lichnowska
Marcin Kozakiewicz

This manuscript is a resubmission of an earlier submission. The following is a list of the peer review reports and author responses from that submission.
Round 1
Reviewer 1 Report
The sample is too limited and not well selected.
The methodology presents many lacks and weaknesses.
The English language is often distracting and poor including orthographic errors.
Thus, in the light of this evidence, it is not appropriate for Applied Science readers.
TITLE
It does not describe the intent of the work.
ABSTRACT
The aim should be better described and above all, it should emerge from the shortcomings found on the subject in the literature
INTRODUCTION
This paragraph does not highlight the objectives of the study well. The literature review is not well developed and, therefore, the background, which led to this research, does not emerge
The MATERIALS AND METHODS section has some lacks:
- More information about the sample recruitment may be specified, such as the occlusal parameters and the cephalometric data.
- The rationale of the sample size should be better. So the lack of sample size pre-determination should be discussed.
- The lack of a control group should be discussed.
- It does not emerge whether an intra-operator and inter-operator analysis, to evaluate the standard error, has been performed.
- No information on examiners emerges from the work: training, age, knowledge of patients' features, etc.
RESULTS
- Results should be better discussed.
DISCUSSION
This section should be better developed. It seems just a repetition of the results, only with a description of the data related to the four examined parameters.
REFERENCES
are not limited to those necessary to support the study. There are some typographical mistakes such as missing volumes or pages and punctuation marks.
The quality of the FIGURE should be improved. the graphs shown are many and repetitive, it would be better to decrease in number and make them clearer
Author Response
Please find my response in attachment.

Reviewer 2 Report
RE: aaplsci-1190660
This article showed an interdisciplinary approach to treat patients with skeletal malocclusion and focused the articulation disorder. This topic is very important in the field of jaw deformity. Thus, this article should be published in this journal after minor revision.
As authors advocated in the discussion section, the type of jaw deformity influences to the articulation, such as open bite and deep bite. In addition, surgical procedures, movement distance, intraoral volume, tongue habit are also influence to the articulation. These analyses should be performed and discussed before acceptance.
Author Response
Please find my response in attachment

Reviewer 3 Report
This paper is investigating an interesting and clinically important topic. And I do believe the whole study was performed properly. However, the data interpretation and presentation must be improved.
- Language needs to be carefully checked and polished. There are some typos and misuses of punctuation marks. There is some improper terminology too. e.g. "palatal/lip cleft" should be "cleft lip/palate".
- In the Methods section, the exclusion criteria (3) doesn't make sense.
- In the Results section, there is a lack of tables or graphs to show some results stated there. e.g. Results stated on Page 6, Line 164-174. Conclusions stated on Page
- Figure 1-6 can be combined into a big and condensed figure. Tables or graphs may be added here to better describe the differences. Question: (1) Are the current figures representative images from a certain patient, or summary of all patients? (2) How to read these figures? What does "density" mean in these figures? Does the x-axis "0-5" represent the 5-point scale?
- In Discussion and Conclusion sections, there is a lack of results to support the conclusions there. e.g. Conclusions stated on Page 9, Line 208-211. Line 226-229. Page 10, Line 265-267.
Author Response
Dear Reviewer,
Here's our reply and modified manuscript. We have elnarged the sample, add a reference group and modified the aim and title.
We hope that the changes and adjustment will please you.
Recenzja 3
- Language needs to be carefully checked and polished. There are some typos and misuses of punctuation marks. There is some improper terminology too. e.g. "palatal/lip cleft" should be "cleft lip/palate". It has been corrected
- In the Methods section, the exclusion criteria (3) doesn't make sense. It has been corrected
- In the Results section, there is a lack of tables or graphs to show some results stated there. e.g. Results stated on Page 6, Line 164-174. Conclusions stated on Page
The whole results and discussion with some conlclusions has been modified.
- Figure 1-6 can be combined into a big and condensed figure. Tables or graphs may be added here to better describe the differences. Question: (1) Are the current figures representative images from a certain patient, or summary of all patients? (2) How to read these figures?
Figures were also modified
- What does "density" mean in these figures? Does the x-axis "0-5" represent the 5-point scale? It was modified, 5 points scale stand for articulation test.
- In Discussion and Conclusion sections, there is a lack of results to support the conclusions there. e.g. Conclusions stated on Page 9, Line 208-211. Line 226-229. Page 10, Line 265-267. Changed and corrected
The new version of the manuscript is in the attachment.
We look forward hearing from You.
Yours sincerely,
Anna Lichnowska
Reviewer 4 Report
Abstract
.04 and .02 are not commonly used p values. Please indicate <.05 in both cases.
Materials and Methods
How sample size calculation was performed?
The scales used for motor skills and pronunciation assessment were already validated?
It would have been interesting to evaluate these data before orthodontic treatment: pre-surgical decompensation could affect speech problems.
Results:
“it was proved that gender, surgery type and speech-language therapy in childhood were not associated with pronunciation improvement.” Appropriate comparisons results should be presented.
A comparison with an untreated group, without orthognathic surgery or without speech therapy (or ideally both control groups), should have been performed to clarify the real effect of the speech therapy.
Class II should have been analyzed separately from class III, and iperdivergent patients separated from hypodivergent patients.
Discussion:
“The same was found between the skeletal open bite and labiodental fricatives” and “a clear link between the misarticulation of fricatives and mandibular overjet was established in the presented paper”: where are presented the data demonstrating these statements?
“Class II and III patients require a maxillofacial surgery to have perfect articulation”: a comparison with a group of adults untreated adults, undergoing speech therapy, must be done.
“The patients affected by any type of prognathism, open bite or retrognathia had lower results in pronouncing fricatives than those without any malocclusion”: there is no comparison with healthy patients, without any type of malocclusion.
Author Response
Dear Reviewer,
We have modified the manuscript according to your notes.
We have enlarged the sample and added the reference group.
We hope that those changes and adjustment will please you.
Here is a detailed reply to your notes:
Abstract
.04 and .02 are not commonly used p values. Please indicate <.05 in both cases. Yest , it is right. It was modified
Materials and Methods
How sample size calculation was performed?
Sample size determination has been not evaluated. Those consecutive patients have been included to experimental group due to limited budget. Instead, we focus on data collection quality: systematic follow-up, cooperation with accredited laboratories ect. If this study is successful in Applied Scienced, then the project manager will argue for a larger budget for further speech studies including sample size calculation.
The scales used for motor skills and pronunciation assessment were already validated?
They are commoly used scales in Poland.
It would have been interesting to evaluate these data before orthodontic treatment: pre-surgical decompensation could affect speech problems.
.Some extra data considering orthodontics was added along with figures and calculations
Results:
“it was proved that gender, surgery type and speech-language therapy in childhood were not associated with pronunciation improvement.” Appropriate comparisons results should be presented.
The whole result part has been changed and reorganized.
We look forward to hearing from you.
Your sincerely,
Anna Lichnowska
Round 2
Reviewer 1 Report
Dear Authors,
many changes have been made to your work, it is now improved and more suitable for the magazine. I ask you to review the English language again and to correct some grammatical errors in the text to perfect it.
Author Response
Dear Reviewer,
We again work hard on the manuscript. We have added the figures and new results about the Skeletal and Soft tissue consonants. We also check and revised English.
We do believe that such an important study will be published soon as it may help to develop new techniques of treating patients by speech therapists.
All changes in the manuscript are those in blue.
We are very grateful for each opinion from you.
Yours sincerely,
Anna Lichnowska

Reviewer 3 Report
- English Language still needs more editing.
- The reference control group is not proper.
- Figures are confusing.
- Some conclusions have no data/result to support.
Author Response
Dear Reviewer,
1. We have again worked hard on the manuscript. We have revised all section and correcting major and minor English mistakes.
2. Answering the note the control group, I would like to kindly point out that over the last 50 years 18 publication on the topic of malocclusion, surgery and speech therapy was published. Only 4 of them had a control group limited in size.
As a speech therapist, I am aware that speech therapy in Poland and other developed countries concerns children, not adults. That is why it is very difficult to collect a control group of adults with speech disorder and without malocclusion. Moreover, most of our patients also visited speech therapists during childhood and it did not help them to improve speech.
Because of this, we added another article to the reference - 25. Hassan, T., Naini, F. B., & Gill, D. S. The Effects of Orthognathic Surgery on Speech: A Review. Journal of Oral and Maxillofacial Surgery, 2007, 65, 12, 2536–2543. doi:10.1016/j.joms.2007.05.018).
3. We have added new figures concerning new results on Skeletal and Soft Tissue Consonants. Those results prove that skeletal malocclusion affects articulation in mandibular prognathism, next laterognathism and only a combined treatment may help to improve speech and affected primary functions. Hopefully, the figures are clear and comprehensive to you.
4. We have corrected conclusions.
All new changes are in blue. We also revised captions and corrected them.
We are grateful for every opinion from you.
Your sincerely,
Anna Lichnowska
Marcin Kozakiewicz